# ADVERSARIAL SHALLOW WATERMARKING

## ABSTRACT

Recent advances in digital watermarking make use of deep neural networks for message embedding and extraction. They typically follow the "encoder-noise layer-decoder"-based architecture. By deliberately establishing a differentiable noise layer to simulate the distortion of the watermarked signal, they jointly train the deep encoder and decoder to fit the noise layer to guarantee robustness. As a result, they are usually weak against unknown distortions that are not used in their training pipeline. In this paper, we propose a novel watermarking framework to resist unknown distortions, namely Adversarial Shallow Watermarking (ASW). ASW utilizes only a shallow decoder that is randomly parameterized and designed to be insensitive to distortions for watermarking extraction. During the watermark embedding, ASW freezes the shallow decoder and adversarially optimizes a host image until its updated version (i.e., the watermarked image) stably triggers the shallow decoder to output the watermark message. During the watermark extraction, it accurately recovers the message from the watermarked image by leveraging the insensitive nature of the shallow decoder against arbitrary distortions. Our ASW is training-free, encoder-free, and noise layer-free. Experiments indicate that the watermarked images created by ASW have strong robustness against various unknown distortions. Compared to the existing "encoder-noise layer-decoder" approaches, ASW achieves comparable results on known distortions and better robustness on unknown distortions. Code is available in the supplementary material. *GPT-5 is adopted to check for grammar, spelling errors, and writing logic issues in the manuscript*

## 1 INTRODUCTION

Digital watermarking aims to embed a piece of message into a certain type of digital media, *e.g.*, images Zhu et al. (2018); Jia et al. (2021), videos Asikuzzaman & Pickering (2017), or audios Bassia et al. (2001), and is one of the main techniques for copyright protection and source tracing. A well-designed digital watermarking algorithm is expected to be both imperceptible and robust. The former requires the watermarked media to be nearly identical to its original version for utility. The latter requires the watermark message to be reliably recovered when the watermarked media undergoes a variety of distortions, which is essential for the algorithm to be applicable in real-world scenarios. Earlier research Van Schyndel et al. (1994) encodes the watermark message by altering the least significant bits (LSB) of the digital media. Later, more studies are carried out in the frequency domain, finding it more robust to embed watermarks in the DCT Ko et al. (2020), DWT Daren et al. (2001), or DFT domains Urvoy et al. (2014) of the digital media.

Like many fields in signal and image processing, digital watermarking is revolutionized by the remarkable development of deep neural networks (DNNs). A typical watermarking paradigm with DNNs is Learning-based Deep Watermarking (LDW) Zhu et al. (2018); Liu et al. (2019); Zhang et al. (2021); Jia et al. (2021); Fang et al. (2023); Tancik et al. (2020); Wengrowski & Dana (2019); Jia et al. (2022); Liu et al. (2023), which follows an autoencoder-like architecture with three basic components: a deep encoder to embed the watermark message into the host image, a noise layer to distort the watermarked image, and a deep decoder to extract the watermark message from the distorted watermarked image (termed as distorted image for short). By deliberately designing the differentiable noise layer to approximate the image distortion, they jointly learn the deep encoder and decoder with the noise layers to guarantee robustness. Despite achieving superior robustness against known distortions simulated by the noise layer, these approaches often fail to resist unknown distortions that are not included in the training pipeline, as shown in Fig. 1.

In this paper, we propose a novel watermarking framework for resisting unknown distortions, namely adversarial shallow watermarking (ASW). Our ASW is training-free, encoder-free, and noise layer-free; it only uses a fixed decoder for watermark embedding and extraction (see Fig. 1). Our insight lies in the fact that, regardless of the type of distortions, it will ultimately manifest as perturbations in the pixel values of the watermarked image. As long as the decoder is sufficiently insensitive to the perturbations of the inputs (i.e., the decoder's output remains unchanged after the perturbation), it is expected to be able to accurately extract the watermark message from the distorted images. In our study, we find that a randomly parameterized shallow neural network is sufficient and appropriate to

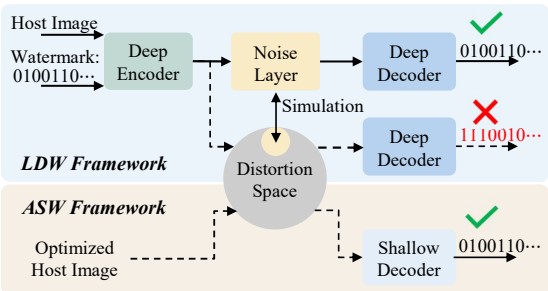

Figure 1: Illustration of the existing LDW framework and the proposed ASW framework, with the solid line representing the case where watermarked images undergo known distortions, and the dashed line representing the case where watermarked images undergo unknown distortions.

be the fixed decoder equipped with the aforementioned property. We denote such a network as a shallow decoder for short in the following discussions. We further propose an adversarial optimization strategy for watermark embedding based on the shallow decoder, which freezes the shallow decoder and iteratively optimizes a host image until its updated version (i.e., the watermarked image) stably triggers the shallow decoder to output the watermark message. In watermark extraction, by using the shallow decoder, the watermark message could be accurately recovered from the distorted image thanks to the insensitivity of the shallow decoder to the image distortions.

Unlike the previous LDW methods that utilize the powerful learning ability of deep networks to fit a set of distortions to ensure specific robustness, our ASW makes use of the insensitive nature of the shallow decoder to achieve resistance against a variety of distortion types. In the experiment, we evaluate our ASW on a dozen of distortion types across a wide range of distortion levels. The results demonstrate that, despite having no prior knowledge of the distortions, the ASW is able to produce high-quality watermarked images with strong robustness against almost all types of distortions. Besides, our ASW demonstrates better robustness than the state-of-the-art (SOTA) LDW methods Zhu et al. (2018); Zhang et al. (2021); Jia et al. (2021); Fang et al. (2023) when the distortions are not seen in their noise layer. The main contributions of this paper are summarized below:

- We conduct empirical studies and analyses to reveal the limitations of the existing LDW methods.

- We propose a novel watermarking framework, ASW, which is training-free, encoder-free, and noise-layer-free, and leverages the insensitivity of shallow networks to guarantee robustness.

- We empirically demonstrate the feasibility of utilizing a single decoder for watermark embedding and extraction, providing a new perspective for future watermark design.

## 2 RELATED WORKS

### 2.1 LEARNING-BASED DEEP WATERMARKING

Recently, learning-based deep watermarking (LDW) methods have been developed, which utilize the powerful fitting capacity of the DNNs and achieve impressive results Zhu et al. (2018); Liu et al. (2019); Zhang et al. (2021); Jia et al. (2021); Fang et al. (2023); Tancik et al. (2020); Wengrowski & Dana (2019); Jia et al. (2022); Liu et al. (2023). LDW typically adopts an "encoder-noise layer-decoder"-based architecture, where the embedding and extraction processes are accomplished separately by the deep encoder and the deep decoder. Zhu *et al.* Zhu et al. (2018) pioneer the research of such a technique, they propose HiDDeN, a LDW scheme capable of resisting several types of image distortions by setting different noise layers. Liu *et al.* Liu et al. (2019) propose a two-stage learning framework for LDW, where the encoder and decoder are trained without the noise layer in

stage one, and the decoder is fine-tuned alone by non-differentiable distortions in stage two. Zhang *et al.* Zhang et al. (2021) find that the main influential component of the noise layer is forward computation rather than the backward propagation. Thus, they propose to replace the back-propagated gradients with an identity transformation. Jia *et al.* Jia et al. (2021) propose a method tailored for resisting JPEG compression, which alternately trains the encoder and decoder using "real JPEG" and "simulated JPEG" noise. In the latest study, Fang *et al.* Fang et al. (2023) use an invertible flow network to achieve watermark embedding and extraction simultaneously, with an invertible noise layer to simulate black-box distortions. Instead of explicitly modeling the distortion layers, Luo *et al.* Luo et al. (2020) utilize the adversarial training strategy Goodfellow et al. (2014) to train a distortion network to generate potential distortions, which has been shown to be effective in resisting unknown distortions. However, it provides inferior performance compared to the SOTA LDW methods Jia et al. (2021); Fang et al. (2023). Additionally, it remains unclear whether the distortion network could model the entire image distortion space.

## 2.2 ADVERSARIAL PERTURBATION

Deep neural networks are sensitive to perturbations. After Szegedy *et al.* Szegedy (2013) discover this intriguing property, many excellent works Goodfellow (2014); Madry (2017); Kurakin et al. (2018); Moosavi-Dezfooli et al. (2016); Carlini & Wagner (2017) have been proposed to generate adversarial examples to fool the network. Some of them generate adversarial examples based on gradients ascending Goodfellow (2014); Madry (2017); Kurakin et al. (2018) using one-step methods for computational efficiency or multi-step methods for more accurate perturbations. Others consider the generation of adversarial examples as an optimization problem Moosavi-Dezfooli et al. (2016); Carlini & Wagner (2017), and take off-the-shelf optimizers Fletcher (2000); Kingma & Ba (2014) to search for the optimal adversarial examples. Adversarial examples have shown to be useful in many applications. Le *et al.* Le Merrer et al. (2020) make use of adversarial examples to protect the copyrights of deep models. Works in Chen et al. (2022); Zhu et al. (2024) utilize adversarial perturbations to prevent valuable datasets from being used without authorization to train deep models. Kishore *et al.* Kishore et al. (2021) apply adversarial examples in image steganography, designing high-capacity, anti-detection steganographic algorithms, where a noise layer has to be incorporated to improve the robustness. In this paper, we propose to take advantage of the adversarial examples in the domain of robust image watermarking, where we attempt to conduct the deep image watermarking by only using a fixed decoder which is capable of resisting general image distortions without any training.

## 3 ANALYSIS OF THE LDW FRAMEWORK

Most of the existing LDW methods train a deep encoder and a deep decoder to fit a fixed set of distortion layers to guarantee robustness. They demonstrate superior robustness against known distortions. However, their robustness is usually unsatisfactory against unknown distortions. In this section, We first analyze such a phenomenon according to the local linear hypothesis Goodfellow (2014), and then provide a theoretical proof of the empirical observation.

### 3.1 EMPIRICAL INVESTIGATION

The local linear hypothesis argues that deep neural networks stack too many linear layers and the popular ReLU activation function Nair & Hinton (2010) runs in a linear fashion. As a result, the error in the input diffuses and magnifies through the linear operations layer by layer, causing a large change to the output. Based on such a hypothesis, we conjecture that the deep decoder adopted in the existing LDW framework is inherently sensitive to perturbations. When their inputs are altered (i.e., watermarked images are distorted), they tend to output results that differ from the watermark. On the other hand, the deep decoder itself has strong learning capability. It fits the known distortions well during the LDW training phase, which makes it insensitive to these distortions. However, for unknown distortions, it remains sensitive and its output would easily be affected due to the change of the input. For justification, we create a binary mask $\mathcal{M}$ with the same size as the host images and use it to generate two mutually orthogonal noise patterns, including $n^+ \sim \mathcal{N}(0, \sigma) \odot \mathcal{M}$ and $n^- \sim \mathcal{N}(0, \sigma) \odot \overline{\mathcal{M}}$, where $\mathcal{N}(0, \sigma)$ represents the Gaussian distribution with mean 0 and variance $\sigma$, $\odot$ is the element-wise product and $\overline{\mathcal{M}}$ is a binary mask complementing $\mathcal{M}$. The cosine similarity

between $n^+$ and $n^-$ is 0, which means the two noise patterns are completely different. We take the $n^+$ and $n^-$ as the known distortion and unknown distortion, where only $n^+$ is used as the distortion layer in the LDW training pipeline.

We conduct the experiments on the popular HiDDeN architecture Zhu et al. (2018), and train several HiDDeN variants by varying the depth of its decoder on the COCO dataset Lin et al. (2014). Then, we evaluate the robustness of the HiDDeN variants to known $n^+$ and unknown $n^-$ distortions on the ImageNet validation dataset Russakovsky et al. (2015). Fig. 2 shows the bit error rate (BER) of the extracted watermark from the decoders of different depths under different distortions. We can see that as the depth of the layer grows, the BER increases against the unknown distortion $n^-$ (i.e., the blue bar), and decreases when considering the known distortion $n^+$ (i.e., the pink bar). The former indicates that the decoder's sensitivity increases as its depth

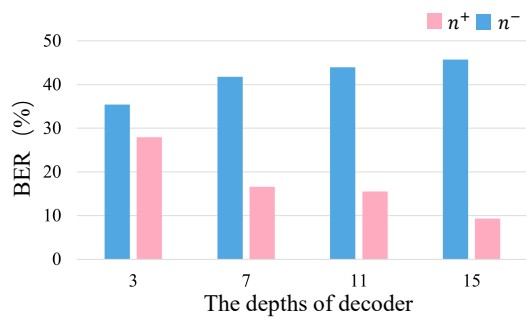

Figure 2: The BER (%) of the extracted watermark message for HiDDeN-decoders of different depths under $n^+$ or $n^-$ distortions.

grows. The latter implies that the decoder's learning ability could effectively compensate for its sensitivity to specific known distortions. This implies that taking learned deep watermarking decoders provide biased robustness against distortions seen in their training, which may not be beneficial for resisting unknown distortions.

## 3.2 PROVABLE SENSITIVITY OF DEEP DECODERS

**Theorem 1** *Consider a decoder consisting of L convolutional layers. When an additive tiny distortion $\delta$ is introduced at the input, the variance of the output error $\Delta y_L$ changes exponentially with the depth L:*

$$\text{Var}[\Delta y_L] = \text{Var}[\delta] \cdot \prod_{l=1}^{L} \left( \frac{1}{2} n_l \text{ Var}[\theta_l] \right), \tag{1}$$

*where $\Delta y_L = y_L - y_L^{'}$ denotes the output error of the network, with $y_L$ and $y_L^{'}$ being the output before and after the distortion, respectively. $\text{Var}[\cdot]$ denotes the variance operator, $n_l$ is the number of filters of the l-th layer, and $\theta_l$ represents the weights of the l-th layer.*

We roughly assume that the expectation of the output error $\Delta y_L$ caused by the unknown distortion $\delta$ is 0, and we approximate the expectation of the squared output error $\Delta y_L$ by the variance as follows:

$$\mathbb{E}\left[ \| y_L - y_L^{'} \|^2 \right] \approx \text{Var}[\delta] \cdot \prod_{l=1}^{L} \left( \frac{1}{2} n_l \text{ Var}[\theta_l] \right). \tag{2}$$

Let $\frac{1}{2} n_l \text{ Var}[\theta_l]$ be denoted as the layer-specific weight variance $v_l$. To solve complex tasks, $v_l$ needs to be relatively larger. For example, the average of $v_l$ in a trained decoder in HiDDeN is 32.34. This suggests that the output error of the HiDDeN decoder grows exponentially with its depth, which is consistent with experimental data in Empirical Investigation. Proofs of Theorem 1 are provided in the Appendix A.

## 4 THE PROPOSED METHOD

Our goal is to develop an image watermarking framework that is robust to arbitrary types of distortions. To this end, we propose adversarial shallow watermarking (ASW). It first establishes a randomly parameterized shallow neural network as the watermark decoder that is insensitive to perturbations of the input. On top of such a shallow decoder, we conduct the watermark embedding by adversarial optimization, where a host image is iteratively updated until its watermarked version stably triggers the shallow decoder to output the watermark message.

## 4.1 THE SHALLOW DECODER

**Weight Setting.** The shallow decoder is expected to provide unbiased robustness against arbitrary distortions. Therefore, it may not be appropriate to use the existing pre-trained deep decoders for setting the weights of our shallow decoder. Such a strategy may construct a decoder that inherits the sensitivity of the existing decoders to some specific types of distortions seen in the training. To deal with such an issue, we propose to randomly set the weights of the shallow decoder. In particular, we adopt a seed $\kappa$ to sample the weights of the shallow decoder (say $\theta$) from the standard Gaussian distribution $\mathcal{N}(0, 1)$, i.e.,

$$\theta = Sample(\mathcal{N}(0, 1), \kappa). \tag{3}$$

**Architecture.** We set the decoder to a shallow neural network stacked with several average pooling (AvgPool) layers, convolutional (Conv) layers, instance normalization (IN) layers, leaky rectified linear units (LeakyReLU), and a full connection (FC) layer, as shown in Fig. 3. Next, we explain why we choose and stack these layers for our shallow decoder.

The purpose of the AvgPool layer (with a stride of 4) is to reduce the decoder's nonlinear response to perturbations in local areas. The weight in each Conv layer is a 4-dimensional (4-D) tensor with the first, second, and last two dimensions being the input channel, output channel, and kernel sizes, respectively. And the strides of all these Conv layers are set as 2. As the weights of the decoder are set randomly, the gradients could have high variance among different samples. We adopt the IN layers after the Conv layers, which do not require the use of the statistics of a mini-batch. After the IN layers, the popular LeakyReLU is adopted to in-

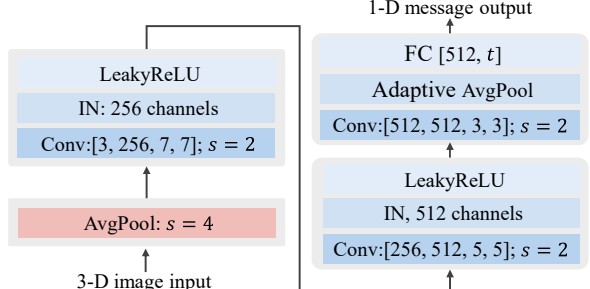

Figure 3: Architecture of the shallow decoder, with $s$ and $t$ representing the stride and output length, respectively.

troduce non-linearity for our shallow decoder. Eventually, an adaptive AvgPool is used to transform the 3-D features from the last Conv layer into a 1-D feature, followed by an FC layer to map the feature values to the watermark message.

## 4.2 ADVERSARIAL WATERMARK EMBEDDING

Let $\mathcal{I}_h \in [0, 1]^d$ denote a RGB host image with $d$ being the number of pixels, and $\mathcal{W} \in \{0, 1\}^t$ be a watermark message to be embedded into $\mathcal{I}_h$ with $t$ being the total number of bits to be hidden. Given a shallow decoder $\mathcal{SD}(\cdot; \theta) : [0, 1]^d \to [0, 1]^t$ parameterized by $\theta$ that takes a 3-D image as input and produces a 1-D message output. The adversarial watermark embedding (AWE) aims to generate a watermarked image $\mathcal{I}_w$, which is close to $\mathcal{I}_h$, to stably trigger the shallow decoder to output the watermark message (i.e., $\mathcal{SD}(\mathcal{I}_w; \theta) = \mathcal{W}$).

Unlike the previous LDW methods that train a deep encoder to create the watermarked images, AWE addresses the problem via an adversarial optimization manner without any knowledge of the distortion, which could be formulated as follows:

$$\min_{\mathcal{I}_w} \mathcal{L}_W + \alpha \mathcal{L}_I$$
$$\text{s.t. } \mathcal{I}_w \in [0, 1]^d \tag{4}$$

where $\mathcal{L}_W$ refers to the watermarking embedding loss to measure the distance between $\mathcal{W}$ and the output of $\mathcal{SD}(\mathcal{I}_w; \theta)$, which is designed by:

$$\mathcal{L}_W = BCE(\mathcal{SD}(\mathcal{I}_w; \theta), \mathcal{W}), \tag{5}$$

where $BCE(\cdot)$ is the binary cross-entropy function. $\mathcal{L}_I$ is the image distortion loss to measure the difference between $\mathcal{I}_h$ and its watermarked version $\mathcal{I}_w$, where

$$\mathcal{L}_I = ||\mathcal{I}_w - \mathcal{I}_h||_2^2. \tag{6}$$

---

**Algorithm 1:** Adversarial Watermark Embedding

---

**Input:** Watermark message $\mathcal{W}$, shallow decoder $\mathcal{SD}(\cdot; \theta)$, seed $\kappa$, host image $\mathcal{I}_h$, number of iterations $iters$, step size $\eta$

**Output:** Watermarked image $\mathcal{I}_w$

1  $\mathcal{I}_w \leftarrow \mathcal{I}_h$
2  $\theta \leftarrow Sample(\mathcal{N}(0,1), \kappa)$                                          ▷ Initialize $\mathcal{SD}'s$ weight
3  **for** $i = 1$ **to** $iters$ **do**
4      $\mathcal{L}_{all} = \mathcal{L}_W + \lambda \mathcal{L}_I$
5      $\mathcal{I}_w \leftarrow$ L-BFGS$(\mathcal{L}_{all}, \eta)$                           ▷ Perturb $\mathcal{I}_w$ to minimize $\mathcal{L}_{all}$
6      $\mathcal{I}_w \leftarrow clip_0^1(\mathcal{I}_w)$;                                ▷ Clip the image to $[0,1]$
7      **if** $\mathcal{H}(\mathcal{SD}(\mathcal{I}_w; \theta)) = \mathcal{W}$ & $\mathcal{L}_{all}$ converges **then**
8         break                                                           ▷ Early stopping
9      **end**
10 **end**

---

$\alpha$ is a hyper-parameter for balancing image quality and watermark decoding accuracy.

The details of our proposed scheme are given in Algorithm 1. We first set $\mathcal{I}_w$ as $\mathcal{I}_h$ and use a seed $\kappa$ to set the weights of the shallow decoder (i.e., $\theta$). We iteratively perturb $\mathcal{I}_w$ to minimize the weighted sum of $\mathcal{L}_W$ and $\mathcal{L}_I$ (say $\mathcal{L}_{all}$). In each iteration, we calculate $\mathcal{L}_{all}$ according to $\mathcal{I}_w$ in the current step. Then, we use the L-BFGS solver Fletcher (2000) to perturb $\mathcal{I}_w$ to minimize $\mathcal{L}_{all}$. The perturbed $\mathcal{I}_w$ is then clipped into the range of $[0,1]$ by the clip function $clip_0^1(\cdot) = \max(\min(\cdot, 1), 0)$.

We perform early stopping when the following two conditions are satisfied: 1) the output of the shallow decoder (after applying the heaviside step function $\mathcal{H}(\cdot)$) equals the watermark message $\mathcal{W}$, and 2) the $\mathcal{L}_{all}$ converges. After the optimization, we enlarge $\mathcal{I}_w$ 255 times to transform it into the range of $[0, 255]$ and quantize it to obtain its RGB version. It should be noted that, in case the image quality of $\mathcal{I}_w$ is poor, we will carry out re-embedding from a random point in the $\epsilon$-ball around $\mathcal{I}_h$, which is given by

$$\mathcal{I}_w \leftarrow \mathcal{I}_h + n \sim \mathcal{U}(-1, 1) \odot \epsilon, \tag{7}$$

where $n$ is a random noise sampled from the standard uniform distribution $\mathcal{U}(-1, 1)$.

In watermark extraction, we can simply feed $\mathcal{I}_w$ to the shallow decoder to extract the watermark by

$$\mathcal{W} = \mathcal{H}(\mathcal{SD}(\mathcal{I}_w; \theta)) \tag{8}$$

## 5 EXPERIMENTS

**Datasets and Settings.**

To evaluate the effectiveness of the proposed ASW, we test it on 1,000 randomly selected images from the ImageNet validation dataset Russakovsky et al. (2015). The width and height of the host image are set to 256, and the length of the watermark message $t$ is set to 36. The hyper-parameter $\alpha$, which balances image quality and watermark accuracy, is fixed at 0.75. In ASW embedding, the seed $\kappa$ is set as a random integer greater than 0, and the number of iterations $iters$ and the step size $\eta$ are set to 25 and 0.05, respectively. The parameter of $\epsilon$ for re-embedding is set to 0.005. The $\sigma$ in Sec.3 is set to $\frac{10}{255}$.

**Benchmarks.** We compare our ASW with several SOTA LDW methods, including HiDDeN Zhu et al. (2018), FASL Zhang et al. (2021), MBRS Jia et al. (2021), FIN Fang et al. (2023) and DADW Luo et al. (2020). To evaluate the robustness, we choose 12 different types of distortions, including JPEG Compression, Gaussian Blur, Median Blur, Gaussian Noise, Poisson Noise, Salt&Pepper Noise, Brightness Shifting, Contrast Shifting, Saturation Shifting, Cropout, Resize, and Rotation. For the Resize and Rotation distortions, we resize/rotate the distorted image back to its original size before conducting the watermark extraction. For a fair comparison, under the same settings for host image size and watermark message length as ours, we retrain the compared methods Zhu et al. (2018); Zhang et al. (2021); Jia et al. (2021); Fang et al. (2023) with their default distortion layers on the MS-COCO dataset Lin et al. (2014). We would like to mention that none of their default

| Methods | PSNR(dB) ↑ | SSIM ↑ | BER(%) ↓ |
|---|---|---|---|
| HiDDeN Zhu et al. (2018) | 36.45 | 0.9674 | 19.31 |
| FASL Zhang et al. (2021) | 26.99 | 0.8956 | 20.08 |
| MBRS Jia et al. (2021) | 39.91 | **0.9825** | **0.00** |
| FIN Fang et al. (2023) | **40.15** | 0.9695 | **0.00** |
| ASW | 38.60 | 0.9707 | **0.00** |

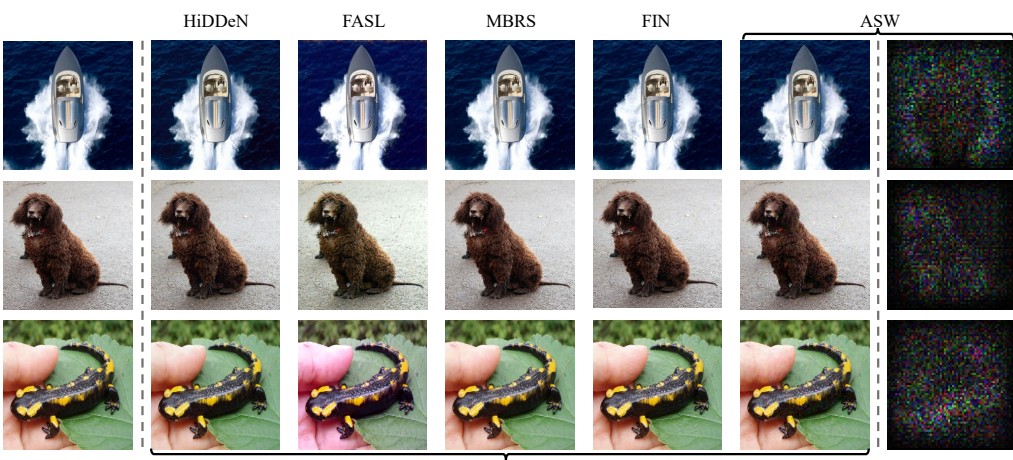

Figure 4: Visualization of the watermarked images generated using the compared LDW methods and the proposed ASW.

Figure 5: Visual quality of the watermarked images and BER (%) of the extracted watermark message from the watermarked images, with the best result in bold. "↑": the larger the better, "↓": the smaller the better.

distortion layers cover all the tested distortions. In other words, there are both known distortions and unknown distortions for them. In contrast, all mentioned distortions are unknown distortions for our ASW. Since the source code of DADW is unavailable, we perform a tailored comparison by evaluating our ASW under the same experimental settings used in DADW Luo et al. (2020) (See Appendix B).

**Evaluation metrics.** There are two metrics adopted to measure the visual quality of the watermarked images, including Peak Signal-to-Noise Ratio (PSNR) and Structural Similarity Index (SSIM) Wang et al. (2004). The larger values of PSNR and SSIM indicate higher image quality. For robustness, we directly utilize the bit error rate (BER) between the extracted and original watermark messages as the evaluation metric, and the smaller BER indicates better robustness. Unless stated otherwise, we report average results on the tested 1,000 images.

The experimental section is organized as follows: Section 5.1 presents the performance of our ASW and the comparison methods when the watermark image is not subjected to any distortion. Section 5.2 demonstrates their robustness against various types of distortions. Section 5.3 compares the computational efficiency of different watermarking methods. Due to space constraints, we place the ablation experiments of our ASW in Appendix C, which investigate ASW's performance when varying its shallow decoder depth, using different initialization methods for parameter initialization, and testing the influence of the AvgPool layer placed in front of our shallow decoder.

## 5.1 VISUAL QUALITY AND BER.

Table 5 presents the numerical results for HiDDeN Zhu et al. (2018), FASL Zhang et al. (2021), MBRS Jia et al. (2021), FIN Fang et al. (2023), and our ASW. We can see that, compared to HiDDeN and FASL, our ASW outperforms them in terms of RNSR and SSIM of the watermarked images, as well as in the BER of the extracted watermark message. All of MBRS, FIN, and our ASW provide reliable watermark extraction accuracy, with all of them being 0.00%. Despite the visual quality of

the watermarked images generated by ASW being slightly inferior to that of MBRS and FIN, the proposed ASW achieves superior robustness when the watermarked image is distorted, which will be discussed in the next section.

Fig. 4 illustrates the watermarked images using different methods. We can see that all of the HiD-DeN, MBRS, FIN, and our ASW create watermarked images with high visual quality. The difference between the host images and their watermarked versions is minimal and almost imperceptible to the human eye. In contrast, FASL's watermarked images contain undesirable color deviation problems. The last column of Fig. 4 shows the magnified residual between the host images and our watermarked images. We can see that, ASW adaptively embeds the watermark message into the host image, where the texture-rich region is embedded with more watermark information. This is beneficial to improve transparency.

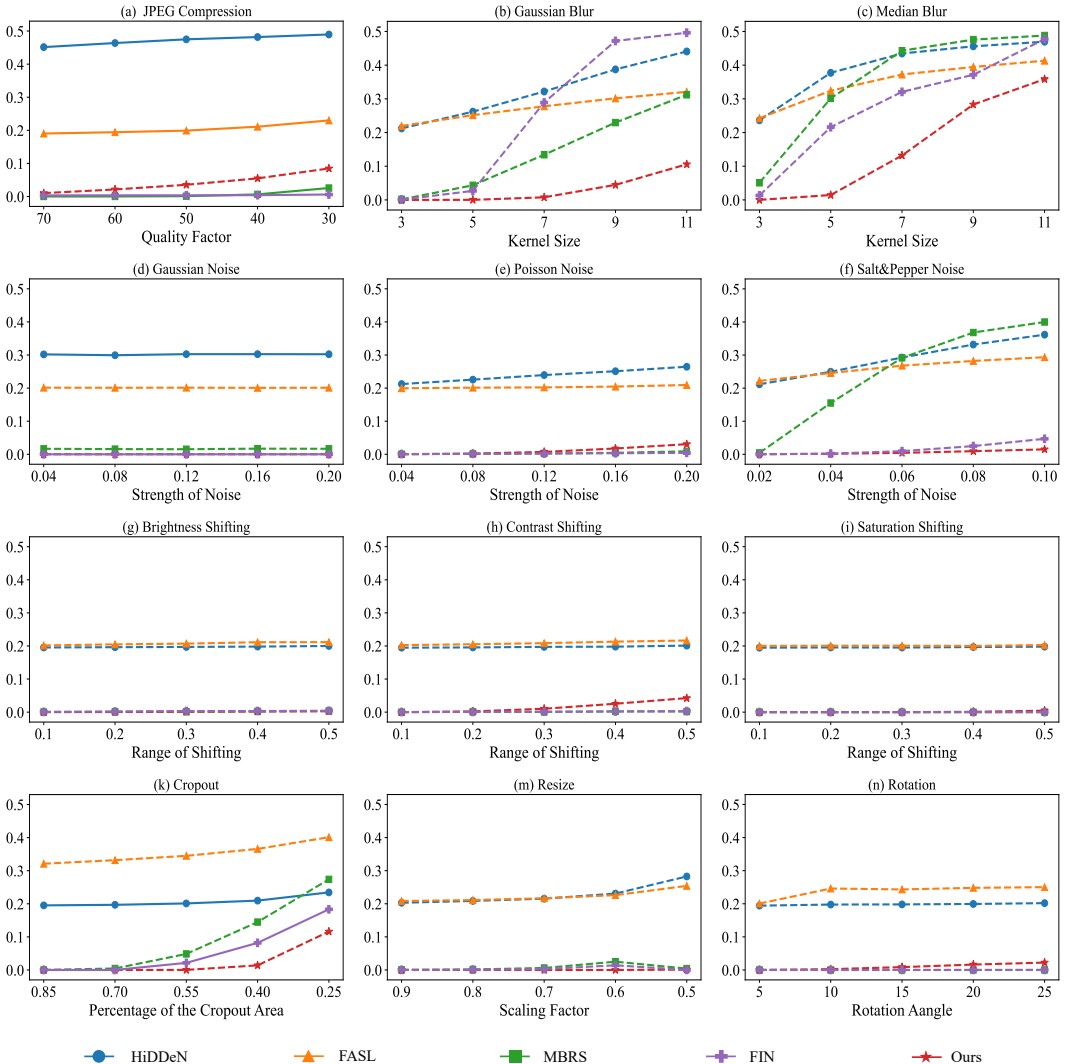

Figure 6: BER (%) of the extracted watermark message for the compared LDW methods and the proposed ASW under different distortions. The solid and dashed lines represent the results of these methods under known and unknown distortions, respectively. The distortion strength increases along the horizontal axis from left to right in all subfigures.

## 5.2 ROBUSTNESS

To evaluate the robustness, we test the compared methods and our ASW on a dozen of distortion types across a wide range of distortion levels and show the results in Fig. 6. It can be seen that ASW

| Methods | Training (h) | Embedding (s) | Extraction (s) | Depths |
|---|---|---|---|---|
| HiDDeN Zhu et al. (2018) | 4.36 | 0.062 | 0.016 | 9 |
| FASL Zhang et al. (2021) | 6.64 | 0.078 | 0.020 | 9 |
| MBRS Jia et al. (2021) | 10.72 | 0.066 | 0.012 | 17 |
| FIN Fang et al. (2023) | 16.85 | 0.072 | 0.072 | 128 |
| ASW | - | 0.887 | 0.002 | 3 |

Table 1: Comparison of computational efficiency with the last two columns presenting the FLOPs and depths of the decoders.

achieves nearly 0% BER against most types of distortions, especially in cases with low distortion levels. which is significantly better than that of HiDDeN Zhu et al. (2018) and FASL. Specifically, we provide a decrease in BER of approximately 20% compared to them across most types of distortions and a wide range of distortion levels.

Regarding resistance to JPEG compression, our ASW achieves favorable results and maintains a BER of less than 10% under JPEG compression with a quality factor of 30. The SOTA MBRS and FIN methods demonstrate impressive robustness and achieve better results than our method. Notably, JPEG compression is simulated and included in their training pipeline.

When encountering unknown distortions (e.g., Gaussian Blur and Median Blur), ASW achieves a lower BER than MBRS and FIN. In particular, under Gaussian Blur with a kernel size of 7, ASW can extract the watermark almost losslessly, while MBRS and FIN have a BER of around 10% and 30%, respectively. A similar result can be observed in the case of resisting Salt & Pepper Noise. Surprisingly, ASW demonstrates the best robustness against Cropout distortion that is a known distortion for HiDDeN and FIN. This indicates the advantages of the proposed method, thanks to the designed shallow decoder, which is insensitive to perturbations.

## 5.3 COMPUTATIONAL EFFICIENCY

LDW framework first jointly trains the deep encoder and deep decoder. Then, it embeds and extracts the watermark message by performing a single forward propagation of the trained encoder and decoder, respectively. In contrast, the proposed ASW framework does not require training networks and uses only a shallow decoder to embed and extract watermarks. It performs dozens of iterations to update the host image for watermark embedding and a single forward pass of the shallow decoder for watermark extraction.

Table 5.1 presents the average computational times for the compared LDW method and our ASW at training, embedding, and extraction stages. We can see that training the deep encoder and decoder of FIN Fang et al. (2023) takes nearly 17 hours (h), which is not long. However, the LDW framework usually requires adjusting the network architecture and retraining the model to accommodate different image resolutions and watermark lengths. This makes the framework time-consuming in real-world applications. Our ASW does not require training networks. Although the embedding time of ASW is longer than that of LDW methods, it takes less than 1.0 seconds (s), which is adequate for use in the majority of real-world scenarios. The extraction time of our ASW is significantly shorter than that of the LDW methods, thanks to our designed shallow decoder with the lowest depths.

## 6 CONCLUSION

In this paper, we propose a novel watermarking framework to resist unknown distortions, namely Adversarial Shallow Watermarking (ASW). Based on the analysis which reveals the limitations of the existing LDW methods, we equip our ASW with a shallow decoder that is randomly parameterized and designed to be insensitive to distortions for watermarking embedding and extraction. ASW conducts the watermark embedding by adversarial optimization, where a host image is iteratively updated until its updated version stably triggers the shallow decoder to output the watermark message. During the watermark extraction, it accurately recovers the message from the distorted image by leveraging the insensitive nature of the shallow decoder against arbitrary distortions. ASW is training-free, encoder-free, and noise layer-free. Extensive experiments have been conducted to demonstrate the advantages of our proposed method for resisting unknown distortions.

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

## A  PROOF OF THEOREM 1

In this part, we investigate the propagation of tiny perturbations through a watermark decoder comprising $L$ convolutional layers. Each layer $l$ is represented by the weight tensor $\theta_l \in \mathbb{R}^{n_l \times n_{l-1} \times k \times k}$, where $n_l$ and $n_{l-1}$ denote the number of filters and the number of channels in the filters, respectively, and $k$ is the kernel size. We assume that the activation function used is ReLU, i.e., $\phi(\cdot) = \max(0, \cdot)$.

We consider a simple case where $k = 1$, and the convolutional layers degenerate into fully connected layers, with the weight matrices $\theta_l \in \mathbb{R}^{n_l \times n_{l-1}}$. Let $\delta \in \mathbb{R}^{n_l}$ represent a small perturbation. The input and output of the $l$-th layer are denoted by $x_l$ and $x_{l+1}$, respectively. The input and output errors at the $l$-th layer, caused by the perturbation $\delta$, are denoted by $\Delta x_l$ and $\Delta x_{l+1}$, respectively. Initially, we set $\Delta x_1 = \delta$. We assume that the direction of error diffusion in each layer is random, and thus, we approximate the expectation $\mathbb{E}[\Delta x_l]$ of the errors as zero.

The output error $\Delta x_{l+1}$ can be expressed as:

$$\Delta x_{l+1} = \phi(\theta_l(x_l + \Delta x_l)) - \phi(\theta_l x_l).$$

For small perturbations, the error can be approximated by the following first-order expansion:

$$\Delta x_{l+1} \approx D_l \cdot \theta_l \cdot \Delta x_l,$$

where $D_l = \mathbf{1}_{\theta_l x_l > 0}$ is the indicator function, denoting the activation state of the neurons. The matrix $D_l$ is a diagonal matrix with entries of 0 or 1, making it an idempotent matrix, i.e., $D_l^T D_l = D_l$. The variance of $\Delta x_{l+1}$ is given by:

$$\mathrm{Var}(\Delta x_{l+1}) = \mathbb{E}\left[ (D_l \theta_l \Delta x_l)^T (D_l \theta_l \Delta x_l) \right] = \mathbb{E}\left[ \Delta x_l^T \theta_l^T D_l \theta_l \Delta x_l \right].$$

We roughly assume that the activation state of the decoder neurons is approximately 0.5, the expected disturbance variance can be approximated as:

$$\mathrm{Var}(\Delta x_{l+1}) \approx \frac{1}{2} n_l \mathbb{E}\left[ \Delta x_l^T \theta_l^T \theta_l \Delta x_l \right].$$

Following previous empirical studies Tian et al. (2021); Huang et al. (2021), it has been shown that the weight distribution of trained neural networks tends to follow a symmetric Gaussian distribution. Thus, we assume that the decoder weights are independent and have zero mean. Therefore, the covariance matrix of the weights satisfies:

$$\mathbb{E}\left[ \theta_l^T \theta_l \right] = n_l \mathrm{Var}(\theta_l) I,$$

where $I$ is the identity matrix. Consequently, the disturbance propagation formula becomes:

$$\mathrm{Var}(\Delta x_{l+1}) = \frac{1}{2} n_l \mathrm{Var}(\theta_l) \cdot \mathrm{Var}(\Delta x_l).$$

After propagating through $L$ layers, the total output disturbance variance is given by:

$$\mathrm{Var}[\Delta y_L] = \mathrm{Var}[\delta] \cdot \prod_{l=1}^{L} \left( \frac{1}{2} n_l \mathrm{Var}[\theta_l] \right).$$

## B  COMPARISON AGAINST DADW

Like ASW, DADW Luo et al. (2020) is designed to resist unknown distortions. The difference is that DADW follows the LDW framework and utilizes an adversarial training strategy to train a distortion network that generates potential distortions. As the source code of DADW remains unavailable, we

evaluate our ASW using the same experimental settings as those employed by DADW Luo et al. (2020). Specifically, a test set containing 3,000 images is randomly selected from the MS-COCO dataset Lin et al. (2014). All images are resized to $128 \times 128$ pixels, with the watermark length $t$ set to 30 bits.

| Metrics | DADW Luo et al. (2020) | ASW |
|---|---|---|
| PSNR (dB) | 33.70 | **37.40** |
| JPEG Compression (50) | 18.30 | **13.63** |
| Gaussian Noise (0.06) | 4.40 | **0.00** |
| Gaussian Noise (0.10) | 10.50 | **0.00** |
| Salt and Pepper (0.05) | 4.30 | **2.71** |
| Salt and Pepper (0.15) | 22.90 | **15.25** |
| Adjust Hue (0.2) | **6.00** | 27.69 |
| Adjust Hue (0.6) | 57.60 | **53.63** |
| Resize Width (0.9) | 0.10 | **0.00** |
| Resize Width (0.5) | 32.90 | **0.00** |

Table 2: Performance comparison of DADW and ASW.

Table 2 compares the performance of DADW with our ASW against various unknown distortions, where the DADW metrics are directly reproduced from the original paper Luo et al. (2020). We can see that ASW achieves higher visual quality, showing a 3.70 dB improvement in PSNR compared to DADW. Furthermore, ASW maintains lower BER in most distortion scenarios, including perfect 0.00% BER under Gaussian noise and image resizing distortions. Notably, DADW exhibits better robustness against the hue adjustment distortion. We attribute this to its adversarially-trained distortion network that better adapts to the distortion space of hue adjustments.

## C  ABLATION STUDY

| Metrics | Depth ($d$) of the decoder ($s = 4$) | | | | Stride ($s$) of AvgPool layer ($d = 3$) | | | |
|---|---|---|---|---|---|---|---|---|
| | $d = 3$ | $d = 4$ | $d = 5$ | $d = 6$ | $s = 1$ | $s = 2$ | $s = 4$ | $s = 8$ |
| PSNR (dB) | 38.60 | 42.57 | 48.12 | 51.59 | 37.89 | 37.97 | 38.60 | 38.47 |
| JPEG Compression (50) | 4.11 | 13.28 | 30.73 | 39.52 | 17.74 | 7.04 | 4.11 | 0.74 |
| Gaussian Blur (9) | 4.78 | 16.50 | 34.80 | 42.66 | 28.47 | 15.95 | 4.78 | 3.20 |
| Poisson Noise (0.2) | 3.24 | 15.08 | 35.14 | 43.96 | 0.14 | 0.50 | 3.24 | 22.03 |
| Contrast Shifting (0.5) | 4.62 | 17.32 | 36.09 | 44.14 | 0.11 | 2.15 | 4.62 | 22.66 |
| Cropput (0.75) | 12.47 | 15.14 | 20.18 | 27.93 | 0.52 | 2.72 | 12.47 | 28.36 |
| Resize (0.7) | 0.00 | 0.02 | 4.50 | 23.02 | 0.41 | 0.00 | 0.00 | 0.00 |

Table 3: Ablation study with the third row displaying the visual quality of the watermarked images, while the following rows report the BER (%) of the extracted watermark message from the distorted images.

**Sensitivity Analysis of Decoder Depths.** We increase the depth of the decoder in Fig. 3 and test ASW's performance. To achieve this, we insert a group of Conv layer, IN layer, and LeakyReLU layer before the last Conv layer of the baseline shallow decoder. The results are presented in Table 4 (columns 2-5). Due to space limitations, we report only a subset of tested distortions. As shown, deeper decoders improve the visual quality of watermarked images, with PSNR increasing from 38.60 dB ($d = 3$) to 51.59 dB ($d = 6$). However , this enhanced depth simultaneously reduces robustness to distortions. For example , BER under JPEG compression (QF=50) rises from 4.11% ($d = 3$) to 39.52% ($d = 6$). This occurs because deeper decoders become more sensitive to perturbations. This sensitivity enables our adversarial watermark embedding (ASE) algorithm to discover smaller perturbations that activate the sensitive decoder to output the watermark message, thereby improving the quality of the watermarked images. Nevertheless , when applying distortions to watermarked images, the sensitive decoder propagates and amplifies noise-caused pixel changes layer by layer, producing mismatched watermarking messages.

**Influence of Initial AvgPool Layer.** We test the influence of the AvgPool layer placed in front of our shallow decoder by varying its stride ($s$) and show the results in Table 4 (columns 6-9), where $s$

= 1 corresponds to the case in which the AvgPool layer is not used. We can see that, larger strides ($s = 4$ or $8$) enhance robustness against structured distortions like JPEG compression (BER drops from 17.74% to 0.74%) and Gaussian blur (BER decreases from 28.47% to 3.20%). However, this comes at the cost of reduced resilience to Poisson noise, Contrast Shifting, and Cropout distortions. To strike a balance, we set $s$ to 4.

**Influence of Weight Initializaion Methods.**

| Initialization Methods | Kaiming | Xavier | Uniform(0,1) | Gaussian(0,1) |
|---|---|---|---|---|
| PSNR ( dB) | 22.21 | 18.99 | 23.53 | 38.60 |
| JPEG (50) | 0.03 | 0.31 | 45.48 | 4.11 |
| Gaussian Blur (9) | 0.02 | 0.03 | 47.00 | 4.78 |
| Poisson Noise (0.2) | 0.02 | 0.09 | 48.91 | 3.24 |

Table 4: Performance of ASW when its shallow decoder is initialized by different methods.

The performance of ASW with its shallow decoder initialized using Kaiming Initialization, Xavier Initialization, Uniform Distribution, and Gaussian Distribution is given in Table 4. The PSNR values for the different initializations are 22.21 dB, 18.99 dB, 23.53 dB, and 38.60 dB, respectively. For BER against JPEG compression at a quality factor of 50, the corresponding values are 0.03%, 0.31%, 45.48%, and 4.11%, while for BER against Gaussian blur, they are 0.02%, 0.03%, 47.00%, and 4.78%. These results show that the Gaussian distribution initialization with a mean of 0 and standard deviation of 1 provides the best trade-off between visual quality and watermark accuracy.

