# OpenReview forum: "Adversarial Shallow Watermarking"
_ICLR.cc/2026/Conference — ICLR 2026 Conference Withdrawn Submission_

### Official Review · Reviewer_f8wQ · 2025-10-17

**Soundness:** 1
**Presentation:** 2
**Contribution:** 1
**Rating:** 2
**Confidence:** 5

**Summary:**

Prior encoder–noise layer–decoder deep watermarking models are robust to training-time distortions but generalize poorly to unknown distortions. To address this, the paper proposes Adversarial Shallow Watermarking (ASW). Unlike prior art, ASW does not require an encoder or a noise layer; and unlike traditional LDW methods that optimize a model via backprop, ASW optimizes the image itself: with a single shallow decoder and adversarial optimization, it produces a watermarked image that stably triggers the decoder. Experiments suggest strong robustness to various unknown distortions.

**Strengths:**

1. Figures are clear; the abstract and introduction are easy to follow.

2. ASW provides a feasible approach to tackle the weakness of deep learning–based watermarking under unknown distortions.

**Weaknesses:**

1.	The decoder-only + adversarial watermarking idea is not new (e.g., LISO [1]), and there is no direct comparison with LISO. LISO reports 3 bpp embedding capacity, while this paper uses 0.00018 bpp. Although robustness is the focus here, the source of robustness in ASW remains unclear. Adversarial examples are typically fragile, yet the adversarially-generated watermarked images here appear highly robust—this contradiction lacks systematic analysis and evidence, which weakens the impact. (The current bpp is too small; please also try the common 64 bits—3×128×128 (~0.0013 bpp) setting used by MBRS/FIN for a fairer comparison.)

2.	Section 3.1 Empirical Investigation uses a highly limited setup that does not properly construct known vs unknown:

    a) M and \overline{M} are complementary. Inevitably, regardless of how n^+ and n^- are sampled, their cosine similarity is 0 due to disjoint supports.

    b) This design is still essentially Gaussian noise (insufficient to represent real-world distortions). With a fixed mask M, the model tends to rely on always-undistorted pixels during training; when tested with the complementary mask, those relied-upon pixels are entirely replaced by distorted pixels, so generalization will collapse.

    c) Given (a) and (b), the setup is too simple to support the current claims.

Additionally, relying only on HiDDeN and only on Gaussian variants is insufficient. Please select some distortions as known and others as unknown for a more faithful evaluation.

3.	Theorem 1 in §3.2 has a very narrow scope (additive noise), whereas the paper targets broad unknown distortions. There is a lack of both theoretical and empirical analysis for non-additive distortions (e.g., geometric, JPEG compression, filtering).

4.	a) The citations in §2.1 contain repetitions/errors and need correction.
b) Figure 5 should be Table 5.

[1] Chen X, Kishore V, Weinberger K Q. Learning iterative neural optimizers for image steganography. ICLR 2022.

**Questions:**

1. What are the exact noise layer settings used for HiDDeN / MBRS / FIN? What are their default noise layers? Did you use single or combined noise layers?

2. The paper focuses on robustness to unknown distortions, and its main way to build robust watermarked images is an adversarial-example procedure. However, adversarial examples are typically fragile, yet the images generated here show broad robustness. The paper does not provide a clear and sufficient explanation or analysis for the source of this robustness.

### Major technical concern: BER under matched settings is significant higher than prior work

**Metric clarification.** In the released code, the variable named `acc` actually computes the **bit error rate (BER)**:
```python
acc = len(torch.nonzero((model(x)>0).float().view(-1) != wm.view(-1))) / wm.numel()
```

This counts the fraction of mismatched bits; thus acc = 0.17 means 17% BER (not 83% accuracy). I report “acc” below as BER for clarity.


Matched configuration vs. MBRS/FIN. I re-ran the authors’ supplementary code, changing only three parameters to match MBRS/FIN: image size = (3,128,128), payload = 64 bits, and weight of criterion_img to 10. All other hyper-parameters were left as in the authors’ script.

My findings (BER, lower is better). Under JPEG q=50, prior work reports very low BER at good image quality:

	•	FIN: PSNR 47.21 dB, BER 0.29%

	•	MBRS: PSNR 42.04 dB, BER 1.35%

With the authors’ code (size & payload changed to match MBRS/FIN), I obtain:

	•	Average PSNR = 33.41 dB (max 38.34 / min 24.44)

	•	JPEG q=50: BER = 17.19%

	•	For reference:

	•	JPEG q=70: 11.51% BER

	•	JPEG q=60: 15.06% BER

Conclusion from reproduction. Even after aligning resolution and payload length to match the baselines, BER remains much higher than MBRS/FIN, and PSNR is ~9–14 dB lower. The gap is substantial and persists across standard distortions.

Why I regard the comparison as not credible/fair.

The method is presented as “robust without training (generating) with a noise layer,” but there is no principled explanation for how adversarially optimizing against a fixed decoder—without modeling the channel—would yield broad robustness. The empirical results from my run do not support such a claim under matched settings.

---

### Official Review · Reviewer_RySq · 2025-10-28

**Soundness:** 3
**Presentation:** 3
**Contribution:** 2
**Rating:** 4
**Confidence:** 3

**Summary:**

This paper introduces ​​Adversarial Shallow Watermarking (ASW)​​, a novel framework designed to address the limitation of existing Learning-based Deep Watermarking (LDW) methods, which exhibit poor robustness against ​​unknown distortions​​ not encountered during training.

**Strengths:**

1. The paper is well written and easy to follow, with a clear and logical structure.

2. The experimental design is comprehensive, and the results effectively demonstrate the validity and robustness of the proposed approach.

**Weaknesses:**

1. The main concern lies in the efficiency of image generation. Traditional methods typically train an encoder–decoder pair, while this paper innovatively proposes to directly optimize the watermark image. However, compared with previous approaches that can rapidly generate a large number of watermarked images, ASW requires significantly more time for image generation, which may limit its practicality. Moreover, the manuscript does not discuss this issue; it only compares training time, which makes the comparison somewhat unfair.

2. The proposed method adopts an optimization strategy similar to adversarial noise generation, but the experiments on distortions do not include evaluations against adversarial defense methods, which would have been important for a complete assessment.

3. Although using a shallow decoder seems to improve robustness against distortions, there is an evident trade-off in terms of imperceptibility. The proposed method does not clearly outperform existing approaches in this regard.

4. The captions of Figure 4 and Figure 5 appear to contain some inconsistencies or potential errors.

**Questions:**

1. Could the authors provide a more detailed analysis of the generation efficiency of ASW compared with conventional encoder–decoder–based watermarking methods?

2. The paper only reports training time, while the actual image generation process might be the more critical factor in practical applications. Could the authors clarify why the comparison focuses solely on training time rather than total processing time (training + generation)? Would including generation efficiency alter the conclusions?

3. The paper shows that using a shallow decoder enhances robustness to distortions, but may reduce imperceptibility. Could the authors elaborate on how they balance this trade-off? Is there a systematic way to adjust network depth or regularization strength to optimize both properties simultaneously?

---

### Official Review · Reviewer_jVTv · 2025-10-30

**Soundness:** 3
**Presentation:** 2
**Contribution:** 2
**Rating:** 4
**Confidence:** 4

**Summary:**

This paper introduces a novel watermarking framework called Adversarial Shallow Watermarking (ASW), designed to resist unknown distortions. Unlike existing deep neural network-based watermarking methods, ASW only uses a shallow, fixed decoder for both watermark embedding and extraction. This decoder is highly robust to perturbations in the input, enabling it to resist various unseen distortions. During watermark embedding, ASW iteratively adjusts the host image through an adversarial optimization strategy until its updated version reliably triggers the shallow decoder to output the watermark. Experimental results show that ASW effectively embeds and extracts the watermark under both known and unknown distortions, outperforming existing deep watermarking methods in terms of robustness.

**Strengths:**

1. The paper presents a novel and original approach to digital watermarking, specifically through the Adversarial Shallow Watermarking (ASW) framework. The originality lies in its departure from traditional deep learning-based watermarking methods, introducing a shallow decoder that is robust against various unknown distortions. This is a significant contribution as it addresses limitations in previous approaches, which often fail to handle unseen distortions effectively.
2. The paper demonstrates clarity and structure reasonably well. The overall structure is clear, and the chapter arrangement effectively communicates the core ideas and experimental results.
3. Regarding data and experimental design, the paper's experimental setup is generally reasonable, and the ASW method is validated through experimental data.

**Weaknesses:**

1. There are writing errors in the paper, particularly in section 5.1, where “Table 5” should be “Figure 5,” and in Table 3, where "Cropput" should be "Cropout." Although these errors do not fully impact the conclusions, they undermine the rigor of the paper, particularly in terms of accuracy in detail.
2. There is an inconsistency in the data presented in the paper. Specifically, in Table 3, the Bit Error Rate (BER) for Cropout 0.75 is 12.47%, but in Figure 6 (k), the data presents a different result. This inconsistency could lead to misunderstandings of the experimental results or cause doubts about the credibility of the paper.
3. Compared to some existing methods (e.g., MBRS and FIN), ASW shows slightly inferior watermark image quality. Particularly when the watermarked image has not undergone any distortion, ASW scores lower in visual quality compared to some deep learning methods.
4. ASW's Performance on Known Distortions (e.g., JPEG Compression):The experiments show that ASW performs worse than some specially trained deep watermarking methods when dealing with known distortions like JPEG compression. While ASW demonstrates strong robustness against unknown distortions, its adaptability to known distortions like JPEG compression is relatively weak.

**Questions:**

1. Does different sampling initialization of the decoder significantly impact the final watermark extraction results?
2. Can you further elaborate on why the shallow decoder is so effective in resisting distortions and successfully decoding watermark information? Is there a theoretical basis or mechanism to explain this phenomenon?
3. Please supplement the performance of the proposed method under real-world distortions, such as screen captures and social media compression.
4. Technically speaking, this article bears a strong resemblance to FNNS[1]. Please outline the similarities and differences between them, along with a performance comparison.

[1] Kishore V, Chen X, Wang Y, et al. Fixed neural network steganography: Train the images, not the network[C]//International conference on learning representations. 2021.

---

### Official Review · Reviewer_WDok · 2025-10-31

**Soundness:** 2
**Presentation:** 2
**Contribution:** 2
**Rating:** 2
**Confidence:** 5

**Summary:**

This paper proposed a novel watermarking framework to resist unknown distortions, namely Adversarial ShallowWatermarking (ASW), which utilizes only a shallow decoder that is randomly parameterized and designed to be insensitive to distortions for watermarking extraction. This method is training-free, encoder-free, and noise layer-free.  The reported experimental results indicate that the watermarked images created by ASW have strong robustness against various unknown distortions.

**Strengths:**

（1）Conduct empirical studies and analyses to reveal the limitations of the existing LDW methods；

（2）Propose a novel watermarking framework, ASW, which is training-free, encoder-free, and noise-layer-free；

（3） Despite having no prior knowledge of the distortions, the ASW is able to produce high-quality watermarked images with strong robustness against almost all types of distortions

（4）Demonstrate the feasibility of utilizing a single decoder for watermark embedding and extraction, providing a new perspective for future watermark design

**Weaknesses:**

（1）The Method section lacks a framework diagram for the entire methodology, which hinders comprehension.
（2）Has the first AvgPool layer in the decoder undergone ablation? What happens if it's removed?
（3）How does the number of layers in a decoder affect decoding accuracy?
（4）The watermark capacity is too small, seriously limiting its practicality.
（5）What would happen if the watermark capacity were increased? (Of course, the host image size could also be expanded accordingly.)
（6）The comparison method is outdated; newer approaches should be introduced.

**Questions:**

See the Weaknesses

---

### Note · Authors · 2025-11-13

**Comment:**

As the first author of the paper, I sincerely thank the Area Chair and the Reviewers of our paper. Given the low rating of my paper, I have decided to withdraw it.

The reviewers were highly professional, providing us with valuable feedback on various aspects, including method design, paper writing, spelling errors, and experimental design. They also pointed out the issues we had avoided with sharp insights.

We will revise our paper according to the reviewers' comments and submit it to another conference/journal.

Finally, I would like to express my gratitude once again to all of you. It is your efforts that have helped improve our paper!

**Withdrawal Confirmation:**

I have read and agree with the venue's withdrawal policy on behalf of myself and my co-authors.